

# Application of multimodal ultrasonography to predicting the acute kidney injury risk of patients with sepsis: artificial intelligence approach

Yidan Tang and Wentao Qin

Emergency Internal Medicine Department, First People's Hospital of Shang Qiu, Shangqiu, Henan, China

## ABSTRACT

The occurrence of acute kidney injury in sepsis represents a common complication in hospitalized and critically injured patients, which is usually associated with an inauspicious prognosis. Thus, additional consequences, for instance, the risk of developing chronic kidney disease, can be coupled with significantly higher mortality. To intervene in advance in high-risk patients, improve poor prognosis, and further enhance the success rate of resuscitation, a diagnostic grading standard of acute kidney injury is employed to quantify. In the article, an artificial intelligence-based multimodal ultrasound imaging technique is conceived by incorporating conventional ultrasound, ultrasonography, and shear wave elastography examination approaches. The acquired focal lesion images in the kidney lumen are mapped into a knowledge map and then injected into feature mining of a multicenter clinical dataset to accomplish risk prediction for the occurrence of acute kidney injury. The clinical decision curve demonstrated that applying the constructed model can help patients whose threshold values range between 0.017 and 0.89 probabilities. Additionally, the metrics of model sensitivity, specificity, accuracy, and area under the curve (AUC) are computed as 67.9%, 82.48%, 76.86%, and 0.692%, respectively, which confirms that multimodal ultrasonography not only improves the diagnostic sensitivity of the constructed model but also dramatically raises the risk prediction capability, thus illustrating that the predictive model possesses promising validity and accuracy metrics.

# INTRODUCTION

Acute kidney injury (AKI) is a common disease with complex etiology, diverse clinical manifestations, high morbidity and mortality rates, and high treatment costs. It can lead to severe conditions and the development of death (*Kunitsu et al., 2022*). The incidence of acute kidney injury in intensive care unit (ICU) is as high as 6.1–70.2%, and in-hospital complication rates, morbidity, and mortality are significantly increased (*Gong et al., 2022*), as well as can lead to more extended hospital stays, additional costs, and poor prognosis. Besides, most patients develop acute kidney injury within 2 days of study enrollment

Corresponding author
Yidan Tang,
tangyidan202211@163.com

(*Molano-Triviño et al., 2022*). Patients with acute kidney injury have a high mortality rate that is often underestimated in clinical practice, as well as increasing length of hospital stay (*Shao et al., 2021*). To a certain extent, the diagnostic criteria of acute kidney injury have deepened the understanding of the probability and the course of acute kidney injury in patients (*Xue et al., 2021*). However, the effects of treatment and intervention in the early stage of identifying the condition vary widely.

The occurrence of acute kidney injury is mainly determined based on changes in functional indicators of serum creatinine (*Hu et al., 2022*). According to studies, serum creatinine is not well sensitive to acute changes in renal function, and levels can vary widely with muscle mass, diet, age, hydration status, medications, and gender (*Zhang et al., 2020*). In addition, it does not act as a marker of renal tubular damage. A marked increase in serum creatinine can be found when renal blood pressure is too low, leading to prerenal azotemia, even if there is no loss of renal tissue. For these reasons, serum creatinine is often defined as the criterion for the deficient criteria for acute kidney injury (*Giretti et al., 2021*). Another problem with serum creatinine is that true baseline values are generally unavailable in most clinical stages. Given the phenotypic variability of acute kidney injury, there is uncertainty about the necessity of different methods to diagnose and monitor the clinical course and treatment (*Lazzareschi et al., 2023*). Meanwhile, urine output is influenced by glomerular filtration rate, diuretics, and other factors. As indicators to judge the occurrence of nephropathy, there are still many shortcomings in both metrics, lacking sensitivity and specificity, failing to reflect early and accurate changes in renal function yet, and consequently leading to delayed diagnosis and treatment.

Several kidney injury biomarkers contribute significantly to early detection of acute kidney injury (*Zeng et al., 2020*). Such novel markers, including kidney injury molecule (KIM-1), are yet to be routinely available for clinical practice (*Wu et al., 2022*; *Zeng et al., 2019*, *2014*, *2013*). Biomarker-based strategies are relatively expensive and have not been appropriately applied owing to the clinical heterogeneity displayed by individual patients (*González et al., 2022*). There is still no effective treatment for acute kidney injury; hence, early identification and active prevention are essential. Therefore, it is necessary to investigate methods to predict potential trends in such disorders to identify high-risk individuals for early prevention.

Acute kidney injury prediction has been a research hotspot in recent years. Several severely ill neonates are divided into three groups (*El-Sadek et al., 2020*). Group 1 has significantly higher mean cystatin C levels on day 1 of incubation. Moreover, serum creatinine and Renal Resistive Index (RRI) are not dramatically different between all groups. At a specific critical value, the area under the curve (AUC) of cystatin C level (0.804) was significantly higher than that of serum creatinine and RRI, which has 53.3% sensitivity and 100% specificity in the early prediction of AKI neonates. RRI possesses a low non-significant AUC at the threshold of 0.53 (0.551), which has 100% sensitivity and 40% specificity, while the other threshold has 33.3% sensitivity and 86.7% specificity. *Zdziechowska et al. (2021)* reviewed the pathophysiology and suggested the best markers to predict contrast-induced acute kidney injury effectively. The variables explored that may predict the development of acute kidney injury on the day of admission for children from

birth to less than 16 years of age admitted to the hospital between 2015 and 2018 (*Raman et al., 2020*). Measurements and primary outcomes showed that the adjusted odds ratio for disease occurrence is 4.2 and the adjusted odds ratio in the non-cardiac cohort is 7.3. Data from spatial cohorts are employed to determine hematological markers (*de Hond et al., 2022*). Cox regression is conducted after dividing the rates into tertiles. One-month mortality, described by the Kaplan-Meier curve, was a significant predictor in Cox analysis. Studies have found that hematological ratios can be used as risk predictors of AKI. In a Thai surgical ICU study, AKI prediction scores are developed using data from critically injured patients undergoing non-cardiothoracic surgery (*Trongtrakul et al., 2020*). As results showed, the model included 3,474 surgically critically injured patients, and 333 (9.6%) developed AKI. This approach functions favorably to predict AKI in critically injured patients undergoing non-cardiothoracic surgery. It is evident that the available prediction method has high feasibility and utility to allow better prediction for the probability of AKI in patients and plays a significant role in preventing AKI.

Sepsis is the most common cause of death in the ICU, and sepsis with AKI has a higher risk of death. Therefore, this article divides the diagnostic criteria for AKI into three grades. Firstly, a multimodal ultrasonography technique consisting of conventional ultrasound, ultrasonography, and shear wave elastography examination method is applied to acquire focal lesion images in the kidney cavity and construct a knowledge map. Secondly, a risk prediction model is built by employing knowledge feature extraction, patient feature representation, and a multicenter discriminator. Thirdly, the development of a tool for risk prediction aims to support clinicians in intervening with patients and reduce the incidence of such, thereby further reducing the progression to end-stage renal disease (ESRD) and mortality in surviving patients.

The rest of the article is structured as follows: Section "A diagnostic criteria for the occurrence of AKI in sepsis" presents the diagnostic criteria for AKI in sepsis. Materials and methods are presented in Section "Materials and Methods". Section "Analysis and Results" presents the analysis and results. Discussion is given in the "Discussion" section. "Conclusions" concludes the research.

## A DIAGNOSTIC CRITERIA FOR THE OCCURRENCE OF AKI IN SEPSIS

AKI is caused by lots of factors, and kidney function can decline within hours. As a consequence, it is often characterized by a rapid decrease in glomerular filtration or accumulation of nitrogenous waste products in the body, which contributes to blood loss or shock in patients with severe sepsis and disorders of water, electrolyte and acid-base balance in the body (*Liu et al., 2020*). Acute kidney injury has a short onset and severe deterioration, often manifesting as clinical complications or syndromes. According to the Global Kidney Disease Prognosis Improvement Organization, acute kidney injury is abnormalities in blood, urine, histology, and imaging (*Ostermann et al., 2020*). The duration of the disease is usually less than three months and the basis for diagnosis is illustrated in Table 1. In this case, the SCR first collected on admission is taken as the baseline value when historical SCR baseline values are unavailable for patients. It can be

**Table 1 Diagnostic criteria for AKI.**

| Standard classification | Index | Details |
|---|---|---|
| Comprehensive standard | SCR | Increase by more than 27.5 μmol/L within 48 h, and 1.8 times or more than baseline within 1 week |
| | Urine output | Less than 0.46 ml/kg * h, the duration is not less than 5 h |
| Staging standard | Phase 1 | SCR | 1.6 times the basal value or increased to 27.1 μmol/L |
| | Urine output | Less than 0.46 ml/kg * h, the duration is 5~10 h |
| | Phase 2 | SCR | 2.3 times the base value |
| | Urine output | Per hour less than 0.46 ml/kg for more than 10 h |
| | Phase 3 | SCR | More than 2.3 times the basal value or increased to 427.1 μmol/L |
| | Urine output | Per hour less than 0.26 ml/kg for more than 1 day or anuria for more than 12 h |

concluded that the grading criteria for acute kidney injury can be defined as grade three. Patients are also included in tertiary staging if treated with RRT or younger than 18 years old with GFR less than 31 ml/min/m$^2$.

Acute kidney injury is treated with fluid therapy, administering vasoactive drugs, controlling infection, and avoiding nephrotoxic drugs. As the indicators measured during the patient's hospitalization are repetitive and varied, some type of variability exists from patient to patient.

# MATERIALS AND METHODS
## Risk prediction model based on multimodal ultrasonography
### Intracavitary image acquisition under multimodal ultrasonography
To provide a basis for risk prediction, this article combines conventional ultrasound, ultrasonography, and shear wave elastography examination methods to acquire images of focal lesions in the renal cavity using multimodal ultrasonography.

### Routine ultrasonographic methods for focal renal lesions
All two-dimensional ultrasound and ultrasonography examinations of focal renal lesions are performed by the same ultrasonographer with more than ten years of experience in renal ultrasound diagnosis. Patients are placed supine with their hands held over their heads to widen the rib cage. The examination was conducted using a Philips EPIQ8 ultrasound C4-1 convex array probe. Patients are fasted for about 5 h and rested for 15 min before the examination. The routine ultrasound examination of the kidneys is carried out in the right subcostal, subxiphoid left subcostal, and right intercostal positions, and in the order of transverse sweep and longitudinal sweep. A comprehensive two-dimensional routine ultrasound examination of the renal tissue is first performed, followed by a focused examination of the detected focal renal lesions (*Minnella et al., 2020*). Furthermore, the location, size, shape, border, margin, interior, and posterior echogenicity of the lesion, presence of calcification, and changes in surrounding tissue structures, blood flow, and spectrum are marked.

### Ultrasonography of kidney focal lesions

After patients sign informed consent, the ultrasonography of focal renal lesions is performed using a Supersonic Imagine Aixplorer ultrasound SC5-2 convex array probe (*Tian, Tian & Jiang, 2021*). The specific operations are presented as follows.

(1) Ultrasound contrast agent administration. A total of 3 mL of saline is drawn and set aside, and then the bottle is shaken to redistribute the microbubbles evenly. A 3 mL supersonic is drawn into a syringe and the mass is injected into a peripheral vein. The spare 6 mL saline mass is injected into the peripheral venous flush line.

(2) After the 2D ultrasound has fully assessed the 2D ultrasound characteristics and color flow spectrum of the focal renal lesion, the largest and most clearly displayed section of the lesion is placed as far as possible in the center of the display while the probe is immobilized. Moreover, the real-time ultrasonography mode is selected, and the image is adjusted to the real-time dual-frame ultrasonography mode. The image depth and gain are adjusted simultaneously to display the best two-dimensional image.

(3) The pre-prepared ultrasound contrast agent is injected into the peripheral vein simultaneously, starting the timer on the screen and launching dynamic video recording, followed by real-time dynamic observation of the ultrasonographic pattern of focal renal lesions.

(4) Dynamic observation and video recording lasted for 5 min.

### Shear-wave elastography of focal renal lesions

The patient rests for 15 min after the ultrasonography examination, and shear-wave elastography is performed with the Supersonic Imagine Aixplorer ultrasound SC5-2 convex array probe (*Chen et al., 2022*). The previous lesion is optimally displayed first, then the elastography mode is selected for elastography examination. Large vessels and bile ducts are avoided as much as possible. The size of sample frames is taken to include as much of the lesion and some of the surrounding renal tissue as possible. If the lesion is large, the sampling frame is placed in the center and at the edge of the lesion (including part of the lesion and part of the surrounding renal tissue). The patient is asked to hold his breath at the end of calm breathing and wait until the color image is as full as possible (larger than one-half of the sampling frame, uniform, no mosaic). The images are accessed when the frame is fixed and stable for 4 s. Young's modulus values of the lesion and surrounding renal tissue are measured and recorded. Young's modulus values for different tissues are measured using the machine's built-in quantitative analysis tools Q-Box and Q-Box Ratio, respectively. The hardness of the tissues in the images is indicated from hard to soft, corresponding to red to blue, with higher Young's modulus values indicating harder tissues. The measurements are performed by using an area of interest of size 15 mm × 20 mm, and the depth should be no greater than 65 mm.

## Risk prediction model for the occurrence of AKI in sepsis

Based on the intracavitary images of the kidney acquired by multimodal ultrasonography, a knowledge-aware risk prediction model is designed, as shown in Fig. 1. The model aims to use the available medical knowledge theory to enrich the semantics of patient
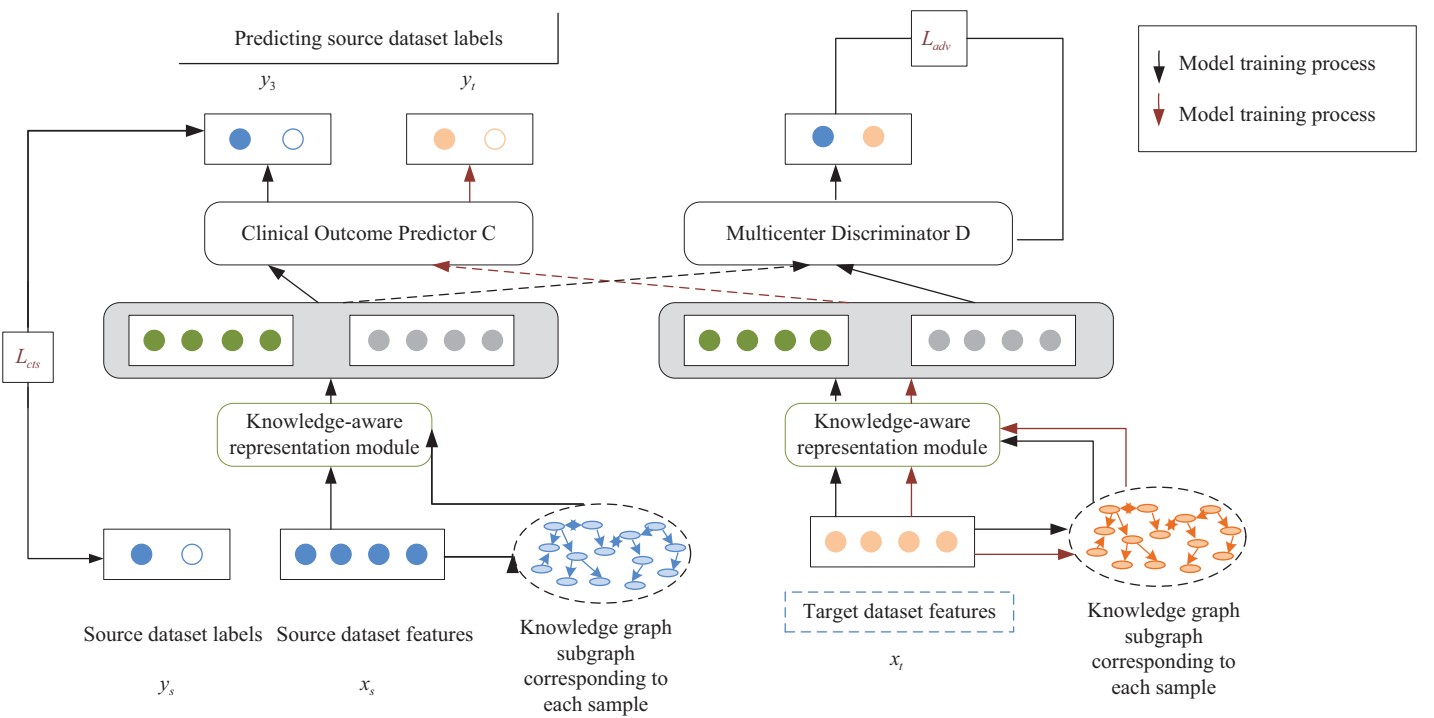

**Figure 1 Schematic diagram of the risk prediction model.**

characteristics and to resolve the problem of poor results of the predicted model caused by differences in the distribution of multicenter clinical data. Suppose the source dataset and the target dataset are denoted by $D_s$ and $D_t$, respectively. The two datasets are from different clinical centers with different marginal data distributions, that is, $P_{D_s}(x) \neq P_{D_t}(x)$. It may lead to a significant reduction in the prediction performance of the constructed model in the target dataset.

The supervised way of the risk prediction model is the unsupervised scheme. The model was first trained with patient data $D_s^l = \{(x_i, y_i)\}_{i=1}^{N_i}$ with the label of AKI occurring in sepsis and patient data $D_t^u = \{(x_i)\}_{i=1}^{N_i}$ without the label as a way to learn information about patient characteristics common to both the source and target datasets.

The proposed risk prediction model injects external knowledge graphs into feature mining of a multicenter clinical dataset. It is also validated by predicting AKI events in unfractionated heparin (UFCH) patients and the Medical Information Mart for Intensive Care (MIMIC-III) patients with sepsis. In this article, learning and extracting features from each clinical center dataset are treated as independent tasks, and standard features of multicenter patients are extracted by using adversarial learning. At the same time, available clinical knowledge from external knowledge graphs is introduced to enrich the semantics of patient features and complete the extraction of knowledge features. Besides, the personality features of patients are extracted by a feature encoder. Finally, the relationship between personality and knowledge features is fully utilized through an attention mechanism based on knowledge perception to predict such occurrences in sepsis.

Specifically, the constructed risk prediction model extracts patient features from specific clinical centers by using a neural network to learn the relationship between the knowledge graph and the original patient features and embeds them into a network model consisting of the following two parts:

(1) A multicenter discriminator that distinguishes patient features of the source and target datasets and extracts common features of multicenter clinical datasets through adversarial learning.

(2) A multilayer perceptron acts as a classifier for clinical outcome prediction. The risk prediction model is divided into three main modules: the knowledge feature extraction, the patient feature representation, and the multicenter discriminator based on adversarial learning. The following section will describe the model construction process from these three modules in detail.

### Knowledge feature extraction

A knowledge map is constructed according to the information obtained from sepsis ultrasonography. Assumed that the constructed knowledge graph is denoted by $g = (\varepsilon, R)$, where $\varepsilon$ refers to the entities in the knowledge graph and $R$ refers to the relationships between the entities in the knowledge graph.

The feature vector is randomly restored, corresponding to the concept $e_i \in \varepsilon$ in each knowledge graph to the initial state. Then, new feature vectors are generated through $l$ layer GCN (*Zeng et al., 2022*). The GCN process is calculated by using Eqs. (1) and (2):

$$h_{e_i} = f\left(\sum_{r \in R} \sum_{j \in N_i^r} \frac{1}{|N_i^r|} W_r^{(1)} g_{e_i} + W_o^{(1)} g_{e_i}\right) \tag{1}$$

$$h_{e_i} = f\left(\sum_{r \in R} \sum_{j \in N_i^r} \frac{1}{|N_i^r|} W_r^{(1)} h_{e_i} + W_o^{(1)} h_{e_i}\right) \tag{2}$$

where $N_i^r$ denotes the set of neighboring nodes of the entity concept $e_i$ in relation $r$. $f(x)$ denotes the ReLU activation function, as shown in Eq. (3). $W_r^{(l)}$ denotes the weight matrix under the relationship at layer $l$, $W_o^{(l)}$ represents the weight matrix of its nodes at layer $l$ and $h_{e_i}^{(l)} \in R^{d_e}$ denotes the feature vector of the concept $e_i$ at layer $l$.

$$f(x) = \begin{cases} 0 & if \ x < 0 \\ x & if \ x \geq 0 \end{cases} \tag{3}$$

After organizing many clinical concepts and transforming them into individual entities, the triad $(e_i, \ r, \ e_j)$ is scored by DistMult factorization. $e_i$ and $e_j$ denote the head and tail nodes, respectively, and $r$ represents the relationship between the two nodes. Equation (4) presents it.

$$s(e_i, \ r, \ e_j) = \sigma\left(h_{e_i}^T R_r h_{e_j}\right) \tag{4}$$

where $\sigma(x)$ denotes the sigmoid activation function $h_{e_i}$. $h_{e_j} \in R^{d_e}$ refers to the eigenvectors corresponding to the clinical concepts $e_i$ and $e_j$, respectively.

For each correct triad $(v) \in K$ and incorrect triad $\left(e_i^r, r, e_j^r\right) \notin K$, the training model has a relationship of scores, namely, $s(e_i, r, e_j) > s\left(e_i', r, e_j'\right)$. Training of the model is to distinguish between correct and incorrect triads, so the following cross-entropy function is chosen as the loss function for the described knowledge feature extraction module:

$$L_g = \frac{-1}{2|\Gamma|} \sum_{\langle(e_i, r, e_j),y\rangle \in \Gamma} (y \log s(e_i, r, e_j) + (1-y) \log(1 - s(e_i, r, e_j))) \tag{5}$$

where $L_g$ represents the loss function of the mentioned knowledge feature extraction module, and $\langle(e_i, r, e_j),y\rangle$ resents the triad with the corresponding labels. When the triad $(e_i, r, e_j)$ is included in the constructed knowledge graph, $y$ takes the value of 1, and otherwise 0. $\Gamma$ represents the set of $\langle(e_i, r, e_j),y\rangle$. After completing the training of the knowledge graph and embedding it into the prediction model of acute kidney injury occurring in sepsis, the feature information in the patient samples $X_{s/t}$ included in the model is obtained by employing knowledge feature extraction.

### Knowledge-aware-based feature representation of patients

The proposed model is composed of a multilayer perceptron neural network with an adversarial learning layer called "a knowledge-aware risk prediction model". After completing the extraction of knowledge features $x^k$, the potential representation of the knowledge feature $z^k$ of the patient is extracted by encoding the knowledge features $x^k$ through the encoder and combined with the decoder to reconstruct. Equation (6) presents it:

$$\begin{cases} z^k = W_{k1}x^k + b_{k1} \\ \hat{x}^k = W_{k2}z^k + b_{k2} \end{cases} \tag{6}$$

where $W_{k1}$ indicates the weight matrix of the encoder, $b_{k1}$ represents the bias of the encoder, $W_{k2}$ denotes the weight matrix of the decoder, $b_{k2}$ shows the bias of the decoder, $z^k$ shows the potential representation of the knowledge feature, and $\hat{x}^k$ refers to the reconstructed feature vector.

In this module, patient samples $x^p$ and knowledge features $x^k$ are used to obtain patient feature representations $z^p$ and knowledge feature representations $z^k$, respectively. Assuming a predictive model decoder of $D_{rec}$, the loss functions of the input $x^k$ and reconstructed feature vectors $\hat{x}^k$ are calculated in Eq. (7):

$$L_{rec}(x^k) = -E_{x_k}\left(\left|\left|D_{rec}(z^k) - x^k\right|\right|_2^2\right) \tag{7}$$

To discover the relationship between patient feature representation $z^p$ and knowledge feature representation $z^k$, an attention mechanism based on knowledge perception was designed. Assuming that the patient feature vector plays a different role in patient performance, the attention mechanism identifies the important features of patients. The attention mechanism is computed in Eq. (8):

$$M = (z^p)^T . z^k \tag{8}$$

where $M$ represents the correlation matrix of the patient sample, representing the dot product of $z^p$ and $z^k$. $M_{i,j}$ represents the relationship between the $i$-th element of the patient feature representation $z^p$ $z^p$ and the $j$-th element of the knowledge feature representation $z^k$.

The normalized feature row vector and the normalized feature column vector, $\alpha^p$ and $\alpha^k$, are calculated by using the SoftMax function for $M$. Equation (9) shows the computational procedure:

$$\begin{cases} \alpha^P = SoftMax\left(\dfrac{\sum_{i=1}^{N} M[.,j]}{N}\right) \\ \alpha^K = SoftMax\left(\dfrac{\sum_{i=1}^{N} M[.,j]}{N}\right). \end{cases} \tag{9}$$

The matrix showing the patients' features of the extracted knowledge is denoted by $B^P$, and the matrix showing the patient-oriented knowledge features is represented by $B^k$. Both are calculated based on Eq. (10), where $U_{p1}$ and $U_{k1}$ denote projection parameters, respectively, $I^P, I^k = [1, \ldots, 1]^T$ denote a $N$-dimensional 1 s vector. $\odot$ refers to the Kronecker product operation, $\odot$ which denotes the multiplication operation between elements.

$$\begin{cases} B^P = \tanh\left(U_{p1}\left(z^P + (I^P \odot \alpha^P) \odot z^k\right)\right) \\ B^k = \tanh\left(U_{k1}\left(z^k + (I^k \odot \alpha^k) \odot z^P\right)\right). \end{cases} \tag{10}$$

After $B^P$ and $B^k$ are obtained, this module generates the feature representation of a patient $\pi^P$ containing knowledge and the feature representation of patient-oriented knowledge $\pi^k$. Equation (11) presents the computational procedure:

$$\begin{cases} \pi^P = B^P.z^P \\ \pi^K = B^K.z^k \end{cases} \tag{11}$$

Ultimately, the feature representation of knowledge-aware patients $\pi(x)$ corresponding to each patient sample $x$ is obtained. Equation (12) presents it:

$$\pi(x) = \left[\pi^P; \pi^k\right]. \tag{12}$$

### Multicenter discriminator based on adversarial learning

This module is an adversarial learning module for extracting common features of patient samples from the source and target datasets, respectively. Specifically, the module is assumed to have a multicenter discriminator $D_{adv}$ that can be used to distinguish patient samples from the source dataset $\pi(x_s)$ and patient samples from the target dataset $\pi(x_t)$. A cross-entropy function is selected as the loss function for the mentioned discriminator. $D_{adv}$. Equation (13) presents it.

$$L_{adv} = -E_{x_s}\left(\log D_{adv}(\pi(x_s))\right) - E_{x_t}\left(\log(1 - D_{adv}(\pi(x_t)))\right). \tag{13}$$

Assuming that the predictor of the clinical outcome of the source dataset is denoted by $c$ the cross-entropy function, the loss function is chosen. Equation (14) presents it.

$$L_{cls} = E_{(x_s, y_s)} = \left( -\sum_{k=1}^{k}(k = y_s) \log C(\pi(x_s)) \right) \tag{14}$$

where $K$ denotes the number of clinical outcome labels. Ultimately, the optimization of the risk prediction model is defined by

$$\Theta^* = arg_{\Theta_{\pi'}} \underbrace{min}_{\Theta_R \Theta_C}(L_{cls} + \lambda_1 L_{adv} + \lambda_2 L_{rec}) \tag{15}$$

where $\lambda_1$ and $\lambda_2$ represent the hyperparameters.

## RESULTS AND ANALYSIS

A model predicting the acute kidney injury risk of patients with sepsis based on multimodal ultrasonography is applied to 100 patients in a municipal hospital. Multiple indicators of clinical decision curve, sensitivity, specificity, accuracy, and AUC are employed to objectively evaluate the validity and accuracy of the constructed model from multiple perspectives.

### Clinical decision curve of risk prediction

A risk prediction model is employed to obtain the probability of acute kidney injury in one of the patients with sepsis. It is defined as positive when the likelihood of occurrence reaches a threshold and interventions are required. There will be benefits for the patient's treatment at this point (*i.e.*, pros). If not treated, the patient with acute kidney injury has a potential chance of losing his life (*i.e.*, cons). On the other hand, a patient with non-acute kidney injury patient does not need any intervention whatsoever. The clinical decision curve shown in Fig. 2 is drawn according to the risk prediction model. The vertical coordinate represents the net benefit after subtracting the disadvantages from the benefits. The bottom horizontal axis denotes the threshold probability. The "ALL" and "None" lines represent the extreme cases. The "ALL" line indicates that if all samples are positive, *i.e.*, all patients with sepsis develop acute kidney injury. All patients receive the intervention; the net benefit is a backslope with a negative slope. The "None" line means that none of the patients have acute kidney injury if all samples are predicted to be negative (*i.e.*, the probability of occurrence is less than the threshold probability). All patients with sepsis do not receive the intervention and the net benefit is zero. The risk prediction curve presents the model's acute kidney injury risk prediction. The farther the curve is from the extremes of the "ALL" and "None" lines, the higher the clinical utility of the proposed model. As shown by the trend of the three curves of the clinical decision graph, the application of the prediction model that resulted in a net benefit for patients ranges between 0.017 and 0.89 threshold probabilities when compared to all patients with and without the intervention treatment. This indicates that the multimodal ultrasonography technique designed in the manuscript improves the validity of the acquired images of focal lesions in the renal cavity by combining conventional ultrasonography methods, ultrasonography methods, and shear wave elastography, thus conferring favorable clinical utility to the proposed model.

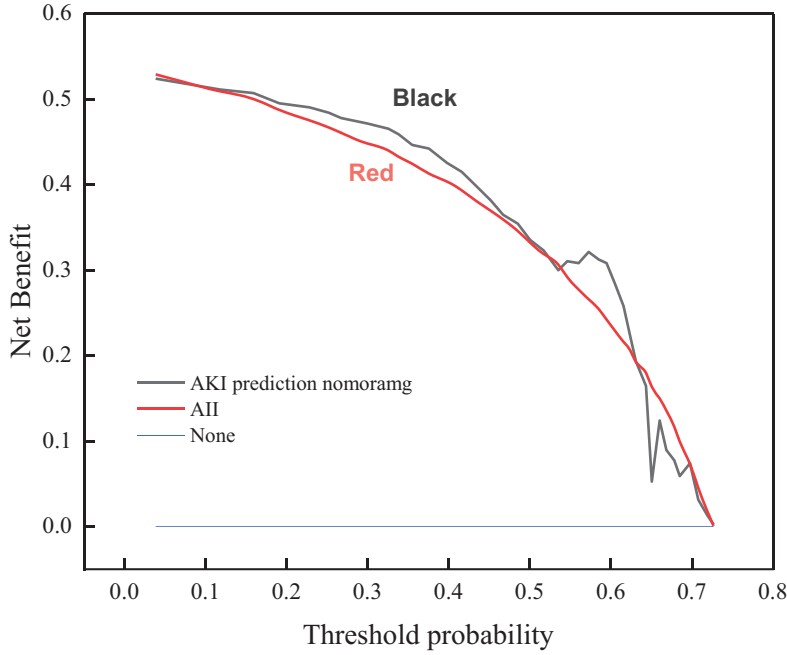

**Figure 2 Clinical decision curve for the risk of acute kidney injury in patients with sepsis.**

## The performance test of the risk prediction model

Five different prediction methods available in the literature are selected to determine the prediction of acute kidney injury: plasma cystatin C and renal resistance index, biochemical indicators, creatinine levels, hematological ratios, and multivariate logistic regression analysis, respectively. The superiority of the proposed method was tested by comparing the prediction results of the five methods. The results of each method and comparison are shown in Fig. 3.

Figure 3 shows that the sensitivity, specificity, accuracy, and AUC metrics had high values of 67.9%, 82.48%, 76.86%, and 0.692, respectively. The implemented method in *Raman et al. (2020)* has the lowest sensitivity of 40.85%. The sensitivity of the other methods was 46.21%, 47.4%, 49.22%, and 42.78%, respectively. The proposed method can predict the risk of acute kidney injury in patients with sepsis. Regarding the two indicators of specificity and accuracy, the method in *Zdziechowska et al. (2021)* has the smallest values, only 64.13% and 61.09%. The other four metrics had specificity values of 66.42%, 68.38%, 67.55%, and 65.29%, and accuracy of 63.78%, 62.05%, 64.27%, and 61.63%, respectively. This shows that the capability of the proposed method to predict the risk of not developing acute kidney injury in patients with sepsis is also relatively strong, while the predictive ability of the other methods is weaker. For accuracy, the proposed method demonstrated superiority relative to the compared methods was 15.77% higher than the lowest value. The AUC evaluation index, which is currently recognized as a better evaluation index, can accurately test the prediction performance of the method. The closer the value is to 1, the higher the authenticity of the method is. If it is equal to 0.5, the authenticity is the lowest and has no application value. After comparing the AUC values of

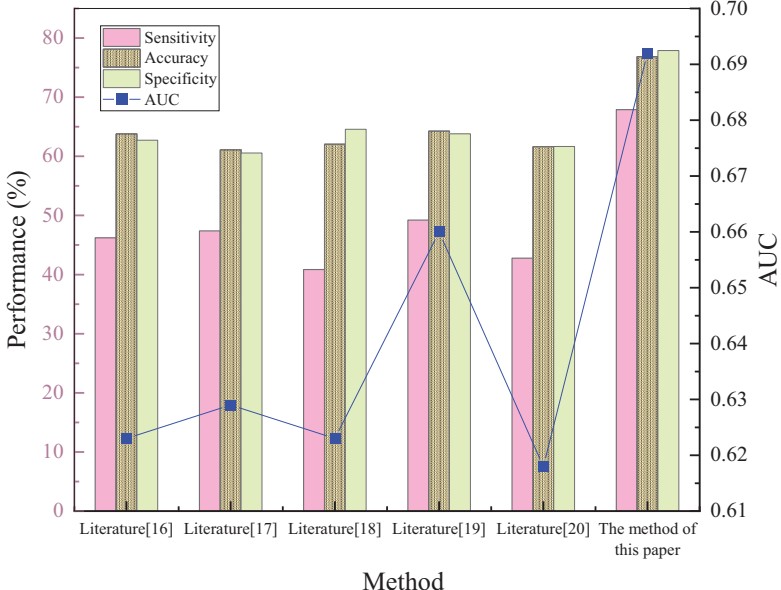

**Figure 3 Schematic diagram of performance validation comparison of different prediction methods.**

the methods, it is found that the AUC values of both the compared methods and the proposed method are greater than 0.5, indicating that each method has some feasibility. The AUC value of the method in *Trongtrakul et al. (2020)* is the smallest, reaching only 0.618, which is much lower than the 0.692 of the proposed approach, indicating significant predictive superiority.

In summary, the manuscript trains the model, learns patient feature information shared with the source, and targets datasets based on intra-cavity images of the kidney acquired by multimodal ultrasonography of patients with sepsis-onset acute kidney injury labels and data of patients without sepsis-onset acute kidney injury label. The knowledge-aware attention-based mechanism fully utilizes the relationship between patient personality traits and knowledge traits. Therefore, the proposed model becomes more favorable to predict acute kidney injury risk when sepsis occurs, which can provide some guidance for preventing and treating acute kidney injury in patients with sepsis.

## DISCUSSION

The following limitations can also be mentioned for the study. Accurate prediction of high-risk patients and timely intervention and preventive measures are also the current hotspot research areas in critical care medicine. The next step of research can be to select some other biomarkers combined with prediction models to predict acute kidney injury risk and further enhance prediction accuracy. The sample in the article consists of only patients in one hospital, is small, and is questionable in its representability. Moreover, external validation is not performed. Various multicenter studies need to be conducted based on expanding the sample size and adding external validation to augment the accuracy of the prediction model.

## CONCLUSIONS

The manuscript uses a multimodal ultrasound imaging technique based on the diagnostic criteria for acute kidney injury in sepsis, combining conventional ultrasound, ultrasonography, and shear wave elastography examination methods. After converting the intra-cavity images of the kidney acquired by this technique into a knowledge map, the risk prediction model of acute kidney injury of patients with sepsis is suggested through feature knowledge extraction, patient feature representation, and multicenter discrimination by adversarial learning. The following conclusion points can be drawn.

(1) The probability of acute kidney injury observed in one of the sepsis patients is obtained by using the risk prediction model, and a clinical decision curve is drawn. The trend of the overall curve shows that the implementation of the prediction model results in a net benefit for the patient, with threshold probability ranging between 0.017 and 0.89 when compared to all patients with and without interventional treatment.

(2) The predictive performance of patients with sepsis is characterized by sensitivity when the risk of acute kidney injury exists. After comparing the sensitivity scores of different methods, it is found that the sensitivity score of the proposed approach is as high as 67.9%, which indicates a solid predictive capability to quantify the development of acute kidney injury risk in patients with sepsis.

(3) Specificity reflects the prediction capability of the patients' risk without acute kidney injury when sepsis occurs. The specificity index score of 82.48% is much higher in the proposed method than in the comparison methods. Therefore, the capability of the constructed approach to predict the risk of sepsis patients without acute kidney injury is also relatively strong.

(4) When the accuracy and AUC evaluation metrics are used to make comparisons, the proposed model demonstrates the same unprecedented superiority and significant feasibility. Thus, the model fully uses the relationship between the patient's personality traits and knowledge traits according to the attention mechanism of knowledge perception. This results in an accuracy of 15.77%, higher than the lowest score and an AUC metric of 0.692.

### Funding

The authors received no funding for this work.

### Competing Interests

The authors declare that they have no competing interests.

### Author Contributions

- Yidan Tang conceived and designed the experiments, performed the experiments, analyzed the data, performed the computation work, prepared figures and/or tables, authored or reviewed drafts of the article, and approved the final draft.

- Wentao Qin conceived and designed the experiments, performed the experiments, analyzed the data, performed the computation work, prepared figures and/or tables, authored or reviewed drafts of the article, and approved the final draft.

## Data Availability

The code is available in the Supplemental File, and the MIMIC-III data is available at https://mimic.mit.edu/docs/gettingstarted.

## Supplemental Information

Supplemental information for this article can be found online at http://dx.doi.org/10.7717/peerj-cs.2157#supplemental-information.

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
