# Peer review of "Application of multimodal ultrasonography to predicting the acute kidney injury risk of patients with sepsis: artificial intelligence approach"

_PeerJ Computer Science, doi:10.7717/peerj-cs.2157_

## Round 0.1 · original submission · Major Revisions

The experts are mainly satisfied with the overall theme and novelty of the paper, but they have a couple of concerns to be addressed before we re-consider your article. Therefore, please carefully revise the paper and resubmit it.
Please ensure to improve the quality of the technical language of the manuscript. thank you

Reviewer 1 ·

Basic reporting

- The occurrence of acute kidney injury in sepsis represents a common complication in hospitalized and critically injured patients. In this paper, the authors proposed multimodal ultrasonography for predicting the risk of acute kidney injury in sepsis. To improve the quality of the paper the authors must incorporate the following given suggestions:

- The author used many abbreviations such as AUC, AKI, RRI in this manuscript but not mention what are their full forms.
- The authors must include a paragraph at the end of introduction which shows the layout of the paper.
- The authors refer the paper in this manuscript as “The method in Ref. [18] has the lowest sensitivity of 40.85%.” It is suggested that the keyword Ref delete from the text. The modified sentence looks like “The method in [18] has the lowest sensitivity of 40.85%.” The same problem occurs many times in paper So, modified them.
- On, line number 63 and 65, the authors start the sentence as “Ref. [19] employed data from spatial cohorts….”. It should be “In [19] the authors employed data from spatial cohorts…”.
- The authors must use the mathematical notations and equation inline.
- The authors include many equations in the manuscript but not refer them. Such as on, line number 170 the authors refer the equation as “The GCN process is calculated by the following equation:” The must change it as “The GCN process is calculated using the Equation 1:”. They also refer the other equation according to the given proposal.

Experimental design

- The figure 1 is vague and the quality is also very poor.
- Both red and black arrows corresponding to Model Training process which shows some ambiguity.
- The discussion given in subsection 4.1 not explaining the results given in figure 2. As the authors mentioned on line 258/259, “As shown by the trend of the three curves of the clinical decision graph, applying the prediction model resulted in a net benefit for patients in the range of 0.017 to 0.89 for the threshold probability fluctuation….” But the figure 2 shows the results against the threshold probability upto 0.72.
- The label of the horizontal line should be Methods instead of Method.

Validity of the findings

- The red line in figure 2 shows the extreme net benefit then how the AKI prediction model produces more benefits than extreme values.

Additional comments

Included in Basic reporting.

Reviewer 2 ·

Basic reporting

The paper introduces an innovative application of artificial intelligence multimodal ultrasonography for predicting the risk of acute kidney injury in sepsis patients. This novel approach combines conventional ultrasound, ultrasonography, and shear wave elastography examination techniques, showcasing the potential of integrating multiple modalities to enhance diagnostic accuracy. However, following changes should be incorporated to uplift the quality of the paper.

Although, the paper provides an overview of various aspects related to AKI diagnosis and prediction, it could benefit from a deeper analysis of the existing literature. Consider elaborating on the methodologies, findings, and limitations of the referenced studies to provide readers with a more comprehensive understanding of the research landscape.

Please proof read the paper carefully to improve the structure of the paper.

The citations provided in the text lack specificity, making it difficult for readers to locate the referenced studies. Ensure that each citation includes sufficient information, such as author names, publication titles, journal names, volume/issue numbers, and page numbers, following a consistent citation style

Experimental design

Please write a critical evaluation of the proposed multimodal ultrasonography technique, including its strengths, limitations, and potential challenges. Discuss factors such as feasibility, scalability, cost-effectiveness, and accessibility to ensure a well-rounded assessment of its practical implications in clinical settings.

Validity of the findings

Connect the findings from the reviewed studies to the proposed multimodal ultrasonography technique more explicitly. Highlight how insights gained from previous research efforts inform the development and potential efficacy of the proposed technique for AKI diagnosis and risk prediction.


Validity and Accuracy: The paper concludes that the predictive model possesses promising validity and accuracy, suggesting its potential as a reliable clinical tool for assessing kidney injury risk in sepsis patients. These findings underscore the robustness of the developed model and its potential for real-world application in clinical settings.

Additional comments

Overall, the paper contributes valuable insights into the application of artificial intelligence multimodal ultrasonography for predicting acute kidney injury risk in sepsis, offering potential benefits for patient care and clinical decision-making.

---

## Round 0.2 · accepted · Accept

Based on the input from Experts, I am pleased to notify you about the acceptance of your article. We thank you for your contribution.

Reviewer 1 ·

Basic reporting

The authors have adequately addressed all the issues raised in my previous report.

Experimental design

The authors have adequately addressed all the issues raised in my previous report.

Validity of the findings

The authors have adequately addressed all the issues raised in my previous report.

Additional comments

The authors have adequately addressed all the issues raised in my previous report.

Reviewer 2 ·

Basic reporting

No comments

Experimental design

No comments

Validity of the findings

No comments

Additional comments

No comments